# Efficient Feature Selection for Static Analysis Vulnerability Prediction

**DOI:** 10.3390/s21041133

**Published:** 2021-02-06

**Authors:** Katarzyna Filus, Paweł Boryszko, Joanna Domańska, Miltiadis Siavvas, Erol Gelenbe

**Affiliations:** 1Institute of Theoretical and Applied Informatics, Polish Academy of Sciences, Baltycka 5, 44-100 Gliwice, Poland; kfilus@iitis.pl (K.F.); pboryszko@iitis.pl (P.B.); seg@iitis.pl (E.G.); 2Information Technologies Institute, Centre for Research & Technology Hellas, 6th km Harilaou-Thermi, 57001 Thessaloniki, Greece; siavvasm@iti.gr

**Keywords:** software vulnerability prediction, static analysis, machine learning, feature selection

## Abstract

Common software vulnerabilities can result in severe security breaches, financial losses, and reputation deterioration and require research effort to improve software security. The acceleration of the software production cycle, limited testing resources, and the lack of security expertise among programmers require the identification of efficient software vulnerability predictors to highlight the system components on which testing should be focused. Although static code analyzers are often used to improve software quality together with machine learning and data mining for software vulnerability prediction, the work regarding the selection and evaluation of different types of relevant vulnerability features is still limited. Thus, in this paper, we examine features generated by SonarQube and CCCC tools, to identify those that can be used for software vulnerability prediction. We investigate the suitability of thirty-three different features to train thirteen distinct machine learning algorithms to design vulnerability predictors and identify the most relevant features that should be used for training. Our evaluation is based on a comprehensive feature selection process based on the correlation analysis of the features, together with four well-known feature selection techniques. Our experiments, using a large publicly available dataset, facilitate the evaluation and result in the identification of small, but efficient sets of features for software vulnerability prediction.

## 1. Introduction

Much effort has been made to avoid security failures in software, which often result from defects in the application source code, known as software vulnerabilities. A vulnerability is a flaw caused by a mistake in the specification, a program, or the configuration of the software. If a vulnerability goes undetected, it will ultimately entail significant maintenance costs [1] and a potential violation of (alleged or explicit) security policies [2]. The term “flaw”, based on the IEEE Standard Glossary of Software Engineering Terminology [3], is the most suitable one to characterize a software vulnerability [4]. While the execution of a faulty section of code does not always violate the security policy, under some conditions, e.g., when specific data reach the faulty code [4], the confidentiality, availability, or integrity of a system may be violated [5].

Security vulnerabilities are often introduced during the coding stage of the Software Development Life Cycle (SDLC), and it is difficult to detect vulnerabilities until they become apparent as security failures in the operational stage of the SDLC because security concerns are not always resolved or known earlier [6].

Thus, it would be of great importance to identify the set of metrics/features that can help to point out the possible occurrence of software vulnerabilities so that testing in the early stages of the SDLC can be used to remove or repair the vulnerability before it becomes apparent and, as a result, allow programmers to consider security from the earliest stages of the development process [7,8].

The detection of software defects after the introduction of a product to the market not only causes the company to have to bear the cost of the repair, but also results in a decrease in the company’s reputation and often entails the expenses of legal proceedings. Therefore, techniques to detect vulnerabilities that can be used in the coding stage of the SDLC are especially valuable [2]. Consequences are even more severe for software vulnerabilities that violate the security and privacy of users and can cause irreparable damage because a majority of users care about data privacy [9]. Indeed, in 2019, personal and corporate data breaches resulted in more than 120 and 50 million dollar losses, respectively [10]. Furthermore, the infamous Equifax data breach caused by the failure of a security vulnerability patch resulted in the exposure of sensitive data concerning 147 million Americans [11]. On the other hand, because of the increase in the number of medical Internet-connected devices and their close interaction with human bodies, new threats to health and life arise, e.g., in January 2017, a software vulnerability created the possibility of gaining control of Internet-connected pacesetters [12]. Such severe consequences of security failures and data breaches resulted in security being identified as “foundational” and a “top IT priority” by the Cisco 2019 Annual Report [13].

To facilitate knowledge about software, security organizations such as the Computer Emergency Response Team Coordination Center (CERT/CC) [14], Open Web Application Security Project (OWASP) [15], and SANSInstitute [16] have been created. These organizations, as well as community and government organizations create public vulnerability repositories (National Vulnerabilities Database (NVD) [17]), vulnerability referencing systems/lists (Common Vulnerabilities and Exposures (CVE) [18], Common Weakness Enumeration (CWE) [19]), rankings (CWE/SANS/25 [20], OWASP Top10 [21]), and guidelines on how to create more secure applications (OWASP Secure Coding Practices Guide [22]). Despite these efforts, vulnerabilities are still common and have severe consequences. It was reported by Veracode [23] that more than 85% of the applications scanned with their security platform (1 April 2017–31 March 2018) contained at least one vulnerability. What is more, in Volume 11 of Veracode’s annual State of Software Security (SOSS) report [24], it was presented that C++ and PHP based applications were the most frequent ones to include high and very high severity flaws. It was 59 percent for C++ and 53 for PHP applications. The acceleration of the software production process, the limited testing resources, and the lack of security knowledge make it impossible to find and fix all of the vulnerabilities and to prevent the resulting exploits.

To prevent security breaches, different techniques can be applied to detect vulnerabilities in source code. Regarding the outputs of the systems, they can be divided into Vulnerability Prediction Systems (VPSs) or vulnerability analysis systems and Vulnerability Discovery Systems (VDSs) [4]. VPSs aim to decide whether a particular part of code (a file, a class, a function, etc.) contains vulnerabilities or not (is vulnerable or neutral). DVSs, on the other hand, target providing more detailed information for particular vulnerabilities found (about a location, a vulnerability type, etc.). No sound and complete system (no missed and no false vulnerabilities), in terms of both the prediction and the discovery, is known to be existent. Therefore, both academic researchers and the software industry put increased focus on the delivery of better and better solutions to facilitate security.

The conventional approaches to vulnerability prediction and discovery can be divided into three groups: static analysis, dynamic analysis, and hybrid approaches [4]. Static analysis (also known as code analysis) is usually conducted during the code review (white-box testing). Many researchers have put efforts into facilitating the performance of static analysis based on many different approaches. However, the derivation and validation of software security properties are still challenging [1]. To perform dynamic analysis, the executable version of the program is necessary. In this type of analysis, the application is scanned during the execution to find vulnerabilities. Because dynamic analysis needs a sufficient number of test cases to find vulnerabilities, it is often very time-consuming [1]. The diverse nature of these two types of analysis makes it a good practice to use both of them during the different stages of the SDLC to increase the probability of creating secure software [25]. Some vulnerabilities simply cannot be detected before the program execution [23], and the others need static analysis to be found. Additionally, static analysis can be introduced in the early stages of the SDLC and examines the whole code of the application, in contrast to dynamic analysis, which focuses on the parts of the executed code. Therefore, it is necessary to incorporate static analysis as a part of software production. The last group, hybrid approaches, uses a mixture of static and dynamic analysis to benefit from these two types, e.g., dynamic analysis is used to eliminate the false positives obtained in static analysis, or static analysis is used to select the test cases for dynamic analysis [4].

Static analysis is a process of system or component examination, which takes into consideration its form, structure, content, or documentation without the code execution [26]. Static analysis tools search for problems in the implementation based on a predefined set of rules, which represent potential anomalies, often occurring in the code. The set of rules consists of a wide range of errors: from mistakes in the source code to complex errors in the system’s logic. Here, we should mention the term, Automatic Static Analysis (ASA), which means that dedicated automated tools are used in the analysis process. ASA alert is in this context, a single report from the ASA tool, which indicates the area in the code, which breaks the predefined static analysis rule. The alert type determines the rule that is broken. The alerts indicate the areas in the source code, in which the execution can be interrupted by, e.g., unverified input data. The types used in [27] were the following: error, mistake, warning, security, and portability. Due to the approximations made during the rule fitting, often a high false positive rate can be observed, and the ASA alerts have to be checked by experts [28]. Code analysis can also provide us with traditional software metrics, which measure some properties of a source code. The examples are: size metrics, complexity metrics [29], complexity, coupling and cohesion (CCC) [30], and also code churn and developer activity [2]. Modern static code analyzers offer us a great variety of different code metrics: traditional metrics (size, complexity, etc.) and a multitude of metrics regarding the number of issues found in the analysis, maintainability, reliability, etc. Software companies often use static analysis tests because they allow eliminating the vulnerabilities even in the coding stage of the SDLC [23]. This type of analysis in its limited form can be performed using Integrated Development Environments (IDEs), e.g., Visual Studio [31] for C/C++, IntelliJ IDEA [32], and Eclipse [33] for Java, as well as special plug-ins created for that purpose. Furthermore, dedicated tools are available on the market: e.g., Veracode [34] and SonarQube [35]. The introduction of vulnerability prediction (usually a binary classification of vulnerable and neutral parts of the source code) enables reducing the number of false alerts to focus the limited testing efforts on potentially vulnerable files [30]. In Section 2, we describe in more detail different approaches to software vulnerability prediction and works related to the topic of our work. Here, also, no sound and complete solution exists [4]; therefore, even more effort should be put into creating more accurate and more efficient solutions. Due to the popularity of the static code analyzers in the industrial world, it is reasonable to use the metrics (traditional ones and those regarding issues, reliability, and dependability) to build software vulnerability prediction models. Therefore, in the current work, we perform a comprehensive analysis of the suitability of these metrics to create ML based software vulnerability predictors and provide some guidelines on what features are most probably the indicators of vulnerabilities. To assess this suitability, we conduct a comprehensive feature analysis and selection (three types of correlation analysis and four feature ranking techniques), as well as an evaluation using thirteen ML models (standard and ensemble ones). The experiments are conducted using a dataset introduced in [36] (available here: [37]), which was based on heterogeneous open-source program files divided into smaller code elements considering resource management error vulnerabilities (CWE-399) and buffer error vulnerabilities (CWE-119) and information gathered from the National Vulnerability Database (NVD) [17] and the NIST Software Assurance Reference Dataset (SARD) project [38]. We generate our features using a commercial tool, SonarQube [35], and a research project, CCCC [39,40].

The remainder of the paper is organized as follows. Section 2 describes different approaches to software vulnerability prediction and related work. Section 3 presents the methodology of the present study: the description of the dataset used in the experiments and feature generation, feature selection methods, and machine learning based evaluation. In Section 4, we describe the results of our experiments. Section 5 concludes the article.

## 2. Related Works

Software vulnerability prediction: Generally, research currently is mainly data-driven and data-dependent. Therefore, machine learning and data mining have gained popularity also in the software vulnerability prediction domain [4]. To predict the vulnerability of a source component, Software Vulnerability Predictors (SVPs) are created. SVPs are highly diversified in terms of the input features, algorithms, and approaches used. Generally, the approaches can be divided into two main groups: calculation based techniques and classification tasks [30]. Calculation based techniques aim to predict a number of vulnerabilities in the system unit, and the classification tasks, the occurrence of vulnerabilities themselves. A unit can be a function, a file, a class, or other component of the system (here, we can also define the granularity of the SVP). In the classification task, software components are labeled as neutral or vulnerable [41]. Different vulnerability types can be treated as one group, or the occurrence of a specific vulnerability type can be detected. The classification approach is the preferable one in the Vulnerability Prediction (VP) domain. SVPs can be based on different types of features: Software Metrics (SM) [2,29,30], Text Mining (TM) [36,41,42,43] features, ASA alerts [27,44], and hybrid ones [45,46,47]. To create SVPs, different algorithms are used: decision trees [43,45], random forests [43,48,49], boosted trees [45], Support Vector Machines (SVM) [50], linear discriminant analysis [2], Bayesian Networks [2], linear regression [45], the naive Bayes classifier [41], K-nearest neighbors [43], as well as artificial neural networks and deep learning [36,47,51,52].

Vulnerability analysis techniques can consider the causes of vulnerabilities, and the others their characteristics and consequences. Some works focus on the consequences of vulnerabilities and risk assessment [53,54]. The detection of the potential occurrence of vulnerabilities in the source code can be used to assess the security risk connected to the product. In [53], known vulnerabilities reported in the National Vulnerability Database along with their complexity, scale, and functionality were used to assess the risk connected to virtual machines. Additionally, knowledge about particular kinds of vulnerabilities can be used to create Intrusion Detection Systems (IDSs) built on the basis of countermeasures to these vulnerabilities (e.g., IDS based on countermeasures to 5G NSAvulnerabilities [54]). On the other hand, it is possible to analyze the causes of vulnerability occurrence and its potential indicators, e.g., features obtained from static analysis.

Many works have been done considering the evaluation of different static code analyzers (e.g., [55] for C/C++), but the number of works considering the analysis of the suitability of features generated by them for the purpose of vulnerability prediction is limited. In [56,57], empirical studies considering three open-source PHP web applications were conducted. They based their research on a dataset and twelve metrics introduced in [49]. In [56], they examined the performance of different software vulnerability prediction models in terms of effort-aware performance measures, in contrast to [57], where they considered the impact of Filter-based Ranking Feature Selection (FRFS) methods on vulnerability prediction. In [58], an empirical study was conducted to examine a security risk (assessed by the Androrisk application) prediction of Android applications based on 21 code metrics obtained using SonarQube and six machine learning algorithms.

In contrast to the related works, we analyze C/C++ applications. C/C++ languages are used in a variety of applications, especially when an interaction at a low level between the application and other components is necessary (a direct interface with the hardware or the operating system), because they offer high control over many aspects and efficiency. These are the languages used to build the majority of operating systems and virtual machines (also the Java Virtual Machine). However, high control and versatility come with a cost, the obligation to avoid bugs and software vulnerabilities, which can entail serious consequences for critical services [59]. According to [24], fifty-nine percent of C++ applications scanned with their analysis tools included high and very high severity flaws. Another aspect can also be observed: high-level programming languages, e.g., Java, are often executed in the dedicated runtime environments provided by virtual machines (usually written in C/C++). Hence, another level of potential vulnerabilities emerges, and software vulnerabilities can occur both in the Java code and in a virtual machine itself. To the best of our knowledge, none of the papers regarding feature analysis for the purpose of vulnerability prediction considered C/C++ programming languages. For that reason, it is crucial to focus on the vulnerability prediction considering lower level programming languages, like C/C++, to facilitate the security of a variety of (often critical) applications, operating systems, and virtual machines, which execute programs written in more abstract languages.

In the current work, we focus on the importance of metrics obtained from static code analyzers for the vulnerability prediction of the C/C++ software components (two types of vulnerabilities classified using Common Weakness Enumeration (CWE) used separately and mixed). We conduct a comprehensive feature analysis and selection, as well as n evaluation using 13 ML models (standard and ensemble ones). We consider three types of correlation analysis and four feature ranking techniques. We use a dataset introduced in [36] (available here: [37]), which was based on heterogeneous open-source program files divided into smaller code elements considering buffer error vulnerabilities (520 files) and management error vulnerabilities (320 files) and information gathered from the National Vulnerability Database (NVD) [17] and the NIST Software Assurance Reference Dataset (SARD) project [38]. We use two tools to generate the features used in the experiments: a commercial tool, SonarQube [35], and a research project, CCCC [39,40]. Although the experiments were conducted on the C/C++ code elements database, the approach can be generalized to other programming languages. For that purpose, it is necessary to utilize a database with code elements written in the language of choice. The other prerequisite is that it is possible to obtain a sufficient number of heterogeneous metrics from a static code analyzer, e.g., SonarQube, or some other alternative, to obtain similar metrics as those described in [60] (but in this case, no guarantee can be made that the features have the same quality as those of SonarQube). SonarQube itself offers static analysis for many different programming languages (e.g., Java, C#, Python, etc.), and we encourage other researchers to conduct similar analyses using different languages, because of the high importance of software security.

## 3. Methodology

In this section, we describe the dataset used in the experiments. We focus on a raw dataset and the process of obtaining the final features used. We also describe the feature selection methods and the evaluation based on multiple machine learning algorithms.

### 3.1. Dataset

Raw dataset: To build the dataset with static code analyzer metrics, it is necessary to gather a sufficient number of code files with the corresponding labels. For that purpose, we used the dataset created in [36] (available here: [37]), which is a set of code components. The dataset contains 61,638 components: 43,913 non-vulnerable and 17,725 vulnerable ones. These files are divided into two groups: the first one considers CWE-119 vulnerabilities (buffer error vulnerabilities) and consists of 10,440 components, and the second one, CWE-399 vulnerabilities (resource management error vulnerabilities), consisting of 7285 components. After labeling the files, we obtained the dataset with 7534 elements in total. The cardinality of the specific subsets is presented in Table 1.

Feature generation: To extract the static code analyzer features, we used two programs: SonarQube [35] and CCCC [39,40]. These tools are widely used in the literature for the purpose of code analysis [61,62,63]. SonarQube is an automatic code review tool. It is provided by a company in Switzerland called SonarSource. They also created the SonarLint extension to some of the most popular IDEs and SonarCloud, which is the cloud implementation of SonarQube. SonarQube allows using multiple languages. However, the static analyzers for some languages (including C/C++) are out of the community version scope. Therefore, we use a plugin [64] to a community version allowing the static code analysis of C and C++ files. The static analysis results are exported to a .csv file, which stores metrics for every file in the dataset. CCCC is a free software tool that was developed by Tim Littlefair. It is a research project that is focused on gathering the software metrics of the program. It provides simple code measurements of the selected file/project. It is used as the Command Line Interface application. By default, the application creates an internal database and the HTML report with the results of the analysis. Since the nature of our database of programs is quite different from the usual application, we had to automate the analysis with a Python script that enabled separate analysis of the files in the dataset. Then, the mirror structure of the folders is created and the summary placed in a corresponding output folder. Then, all of the analyzed results are gathered into one .csv file. We connect both of the .csv files with the labels and achieve the datasets, which are used in the analysis. There are three datasets: one considering only CWE-399 vulnerabilities, the second one, CWE-119 vulnerabilities, and the last one, both of them. Each of the final datasets consists of 33 heterogeneous features.

### 3.2. Feature Selection

High-dimensional attribute sets can contain irrelevant features, which introduce additional “noise” and difficulty for the learning algorithm because the meaningful information has to be extracted from the multidimensional feature space. Reducing the dimensionality of the features can decrease the execution time of the learning algorithm, and a good feature selection can improve the performance of the final model. Furthermore, using feature selection techniques can reduce overfitting and, as a result, contribute to the better generalization of the model.

To determine the quality of the features, we used three types of commonly used correlation analysis techniques (Pearson correlation [65], Spearman correlation [66,67], and Kendall correlation [67]), three types of entropy based ranking methods (information gain, information gain ratio, and Gini decrease index), and the $\chi^{2}$ ranking technique (the descriptions can be found later). The purpose of correlation analysis in this work is to detect whether there are features that demonstrate a statistically significant correlation with the class attribute. For that purpose, after calculating the correlation coefficients’ values, we performed significance analysis. A similar approach was followed in [68], where only the Pearson, Spearman, and Kendall correlation, mutual information (information gain), and $\chi^{2}$ ranking technique were used. All of these methods can be used for ranking the features for the purpose of feature selection (choosing features with the best rank values for a particular method); however, their nature and purpose are different.

Correlation techniques are commonly used to examine relationships between variables, preferably the continuous ones. The correlation coefficients measure different relationship types. Spearman’s or Kendall’s correlation coefficients indicate the occurrence of monotonic relationships (they do not have to be linear) between the two variables in contrast to Pearson correlation coefficients, which determine only the linear relationship. The Pearson correlation coefficient is a parametric measure, and Spearman’s or Kendall’s correlation coefficients are non-parametric [69]. It is possible to treat ordinal variables as continuous variables, but this can introduce potentially incorrect estimation of correlational measures, especially when there are few ordinal categories. This problem refers mainly to the Pearson correlation, which should not be applied to ordinal variables [70]. Nevertheless, the approaches based on the Pearson correlation are robust and can often successfully find a linear association even when the traditional assumption is violated [71]. For that reason, we decided to show the results of all the correlation coefficients, but to highlight that in the case of ordinal variables (especially ones with a small number of values), it is safer to use one of the rank correlation coefficients (Spearman or Kendall) and to use only Spearman correlation coefficient values to select the best set of features obtained in the correlation analysis.

Entropy based techniques and the $\chi^{2}$ ranking technique, on the other hand, can be used when dealing with all types of features, and they are based on the statistical properties of the variables. These methods are commonly used in the feature selection domain [68]. Nevertheless, these methods are not perfect either, e.g., the $\chi^{2}$ ranking technique does not perform well while dealing with infrequent terms in data [72], and information gain favors features with many uniformly distributed values [73]. For that reason, it is a good practice to test different types of feature selection and to perform the additional evaluation, e.g., machine learning based evaluation, which was also done in [68] and in our work (described in the next subsection).

We used the aforementioned commonly used correlation methods to obtain the correlation coefficients values and the corresponding *p*-values. Then, to test the significance of the correlation coefficient, we performed the hypothesis test using the *p*-value. We define the hypotheses as follows:

**Hypothesis** **1** **(H1).**The correlation between the particular feature and the label value is significant.

**Hypothesis** **2** **(H2).**The correlation between the particular feature and the label value is not significant.

We used the level of confidence α=0.05. By comparing the *p*-value with α, we could conclude if the null hypothesis H1 should be rejected. We rejected the null hypothesis if the *p*-value was not less than the significance level, and we did not reject the hypothesis if the value was less than α.

Information gain is a statistical property originally used to select the attributes used in the succeeding nodes of the decision tree in the ID3algorithm [73]. The information gain determines the efficiency of using a particular feature to separate the samples according to their class membership. The measure is based on entropy, one of the basic terms in information theory. Entropy for a binary classification can be defined as follows:(1)$$Entropy\left(S\right)=-p_{1}log_{2}p_{1}-p_{0}log_{2}p_{0},$$
where $p_{0}$ and $p_{1}$ are the probabilities of a sample being a member of a negative and a positive class, respectively. In the general case, for the multi-class classification, this transforms to:(2)$$Entropy\left(S\right)=-\sum_{i=1}^{N}p_{i}log_{2}p_{i},$$
where $p_{i}$ is the probability of a sample being a member of an *i*-th class. Now, information gain can be defined as follows:(3)$$InfoGain\left(S,A\right)=Entropy\left(S\right)-\sum_{v\in V_{A}}\frac{|S_{v}|}{\left|S\right|}Entropy\left(Sv\right),$$
where $V_{A}$ is the set of possible values of a feature A and $S_{v}$ is a subset of $V_{A}$, for which the feature A takes the value v. With such a defined measure, we can clearly see that the measure describes the reduction in the entropy, which is expected when the examined attribute is used to divide the dataset according to classes [73].

Gain ratio is another technique to choose the decision attributes. It penalizes the features with many uniformly distributed values. It uses the earlier defined information gain measure, as well as the measure defined as follows [73]:(4)$$SplitInfo\left(S,A\right)=-\sum_{i=1}^{N}\frac{|S_{i}|}{\left|S\right|}log_{2}\frac{|S_{i}|}{\left|S\right|},$$
where $S_{i}$ is the subset of instances, a result of partitioning the set *S* according to *N* classes. The metric measures the entropy of S, but takes into consideration the number of distinctive values of the observed feature. The gain ratio can be defined as follows [73]:(5)$$GainRatio\left(S,A\right)=\frac{InfoGain(S,A)}{SplitInfo(S,A)}$$

The Gini decrease or the decrease in the Gini impurity can be interpreted as the decrease of the impurity between the subsequent nodes in the random forest [74]. Impurity determines the probability of obtaining two instances of the separate classes in two draws, with the assumption that the distribution of instances is multinomial. The examples from the dataset (it can be perceived as a node using the random tree terminology, *w*) are divided into two parts (into two child nodes, $w_{1}$ and $w_{2})$. The decrease in Gini impurity can be evaluated as follows [74]:(6)$$\Delta i\left(w;v;\eta_{v}\right)=i\left(W\right)-\frac{n_{w_{1}}}{n_{w}}i\left(w_{1}\right)-\frac{n_{w_{2}}}{n_{w}}i\left(w_{2}\right),$$
where *v* is the considered feature and η is the threshold for this value. The features are selected to maximize the reduction in the impurity.

The $\chi^{2}$ technique measures the difference between the observed number of instances of feature *f* for a particular class from the expected value. It is assumed that no feature is dependent on the label *c*. The measure can be defined as follows [75]:(7)$$\chi^{2}\left(f\right)=\sum_{c=1}^{k}\sum_{i=1}^{m}\frac{{\left(O\left(f=v_{i},c\right)-E\left(f=v_{i},c\right)\right)}^{2}}{E(f=v_{i},c)},$$
where $v_{i}$ is the category, *k* the number of classes, and *m* the number of instances of the feature *f*. $O(f=v_{i},c)$ is the number of $v_{i}$ instances in the feature *f* with value *c*. It can be used in the random variable independence test. Let us define the zero hypothesis as H1 (The random variables are independent) and the alternative one H2 (the examined variables are not independent). The bigger the value of the statistic $\chi^{2}\left(f\right)$, the bigger the chance that the examined random variable is correlated with the decision class and the null hypothesis should be rejected [75].

### 3.3. ML Based Evaluation

Using multiple heterogeneous machine learning models in a supervised task and observing their performance can be treated as an indicator of the “worthiness” of the training dataset (e.g., different types of vulnerabilities). Acceptable results of the majority of algorithms can be an indicator of the reliability of the features used [76]. Selection methods used in the study are different in nature and produce different subsets of features as “best features”. That is why we also used machine learning models to test the effectiveness of the feature selection methods in the case of features generated from a static code analyzer.

To evaluate the performance of different machine learning models, we used 5-fold cross-validation. We used thirteen different ML algorithms and eight standard ones: decision trees, k-Nearest Neighbors (KNNs), logistic regression, kernel naive Bayes, SVM (linear, quadratic, cubic, and Gaussian kernels), and five ensemble models: boosted trees, bagged trees, subspace discriminant, subspace KNN, and RUSBoostedtrees. As the model’s performance indicators, we used three standard ML metrics: accuracy, recall (sensitivity, True Positive Rate (TPR)), and specificity (True Negative Rate (TN)R). Accuracy gives an overall evaluation of the model’s performance, and the two additional metrics focus on particular parts of the prediction: the correctness of the prediction of the vulnerable elements (recall, sensitivity) and the neutral elements (specificity) [77].

Some of the ML models, so-called white-box models, allow us to obtain the interpretation of the prediction process and, as a result, an even more detailed evaluation of the features’ reliability. In our work, we decided to present the graphs of decision trees trained on two subsets of data, the first one with only CWE-119 vulnerabilities considered and the second one with CWE-399 vulnerabilities. We used all 33 features to let the model decide what features it uses to make the decision. This knowledge can be used by experts to evaluate the security of code elements (or at least give them some insights into the issues that are chosen by the models as the strongest indicators of vulnerabilities).

## 4. Experimental Results

In Figure 1, Figure 2 and Figure 3, we present the correlation coefficients for different coefficients and subsets of the dataset, considering both types of vulnerabilities, and also specific vulnerability: CWE-399 or CWE-119. We clearly see that among the coefficients, the highest values are reached for the CWE-399 subset. In the majority of cases, the correlations are negative, and their strength is weak to moderate. We can also observe that the correlation coefficients’ values are similar for two rank correlation coefficients, Kendall’s and Spearman’s, and differ significantly for Pearson’s coefficient. This can be caused by the fact that some of the metrics used in this study are ordinal values with a small number of values and that the Pearson correlation can only determine the linear relationship between the variables.

To examine the significance of the correlations, we used the level of confidence α=0.05 and determined whether the null hypothesis H1 (stated in Section 3.2) should be rejected. In Table 2, we give the *p*-value results for the columns, in which it is larger than α. In other cases, the *p*-values were ≪0.001. The underlined, bolded values are the values that suggest that the null hypothesis H1 should be rejected. In these cases, the conclusion is that there is insufficient evidence that the correlation between the particular feature for this type of correlation is significant. In the group of features for which the null hypothesis was rejected at least in one case, there are three features obtained from the CCCC analyzer.

To build the 10 best features dataset using the correlation results, we used the features with the highest value of the Spearman correlation coefficient on the dataset with both types of vulnerabilities considered. For this case, the *p*-value is more than the significance level only for the feature *CCCCMVGCOM*. This feature is not included in the reduced dataset.

We ranked the features using the information gain, gain ratio, Gini decrease, and $\chi^{2}$ technique to determine the ten best features for each of the methods, which we then used to train the thirteen ML models. We chose the ten best features in the sense of the whole dataset to standardize the evaluation process. The results of this analysis can be seen in Figure 4. The features used in the reduced sets are listed in Table 3. We can notice that seven out of ten features are the same for all of the subsets. These features are: code_smells, open_issues, violations, major_violations, sqale_index, comment_lines_density, and critical_violations. Most of them are connected to the number of issues found in the code by SonarQube, which seem to be reasonable choices to build the vulnerability prediction model. What is crucial is that none of the pairs of subsets contain the same information.

Additionally, we performed the same analysis on the subsets of the dataset focused on the CWE-399 (Figure 5) and CWE-119 (Figure 6) vulnerabilities. Because of the higher number of samples from the CWE-119 subset, the overall values of the metrics are influenced more by these samples. The 10 best features according to the information gain for the summary dataset consist of nine out of the ten best features from the CWE-119 subset. One feature, minor violations, comes from the other subset, and it has the highest value of the information gain for the CWE-399 subset. Seven out of these 10 best values can be found in the group with the relatively high information gain values for the CWE-399 subset (not in the first ten values, but the values ranked 4 to 19 are all on a similar level of the information gain value). For the gain ratio, eight out of the 10 best features overall were found in the CWE-119 10-best features, and seven out of the 10 best features overall were found in the CWE-399 10 best features and the relatively high subsequent values. Similarly for the Gini decrease, there were 9/10 features for the CWE-119 subset and 7/10 features for the CWE-399 subset. For the $\chi^{2}$ metrics, the ratios were 8/10 for the CWE-119 subset and 9/10 features for the CWE-399 subset. For CWE-399, the values of all metrics decreased slower than in other cases, which is why it is reasonable to also consider some of the features not included in the 10 best ones as potential information sources for the model.

We aggregated the performance results using the different subsets of the input features as the grouping attribute and the following methods to obtain the aggregation: mean value, standard deviation, maximum value, and minimum value. The results can be observed in Figure 7 (accuracy), Figure 8 (recall), and Figure 9 (specificity). Figure 7 shows that the average accuracy results are comparable for all the subsets of input features. The standard deviation is also comparable for all the examined cases. We can observe a significant difference between the maximum results for the subset considering only CWE-399 vulnerabilities, which suggests that these types of vulnerabilities are easier to detect using the features from a static code analyzer. For the minimum values, this difference is less visible. The highest minimum value was obtained for the CWE-399 based subset with all features included. The results were much more varied for the recall and specificity, more for recall than for specificity. In Figure 8 and Figure 9, it is visible that the minimum results of the specificity are in the majority of cases higher than the recall ones. In almost all cases, the highest average results were obtained for the full set of features. This dataset is the biggest one, contains the most varied types of information, and delivers the most general information, which can be used by different classifiers.

We also aggregated the results by grouping them by the type of ML model to assess the performance of the models trained on different subsets of the dataset. Again, we used three metrics and their statistical features (mean, standard deviation, maximum, and minimum), namely accuracy, recall, and specificity. The results can be seen in Figure 10, Figure 11 and Figure 12.

In Figure 10, it is visible that in terms of accuracy, the classifiers with the highest performance are the KNN algorithm and bagged trees. Their performance is also good in terms of the maximum and minimum value, as well as the standard deviation, that is 6% for both models. The models that obtained the worst results were logistic regression, naive Bayes, SVMs (the exception was the SVM with the Gaussian kernel), and subspace discriminant.

In Figure 11, we can clearly see that the recall results are much more varied than the accuracy ones. Although the maximum values of the classifiers take similar values, the minimum values and the average show that some of the classifiers are highly unstable in terms of recall, namely naive Bayes and SVMs with cubic and quadratic kernels. For these classifiers, the minimum value of recall is close to 0%, which means virtually no predictions of vulnerabilities.

In terms of specificity (Figure 12), the results are more reliable than the recall ones; however, in the case of SVM with the cubic kernel, the standard deviation value reaches 31.18% and the minimal value 0%. Furthermore, the SVM with the quadratic kernel is highly unreliable.

The ensemble models achieve reliable results in terms of all the ML metrics and the grouping categories (avg, std, max, min), besides the subspace discriminant.

Additionally, we present detailed information about all the performance results gathered in the study. They can be observed in Figure 13 (accuracy), Figure 14 (recall), and Figure 15 (specificity). The highest performance of the models was achieved for the subset of data considering only CWE-399 vulnerabilities using the KNN algorithm and the bagged trees. In the case of bagged trees, the best models in this category reached 94.4% accuracy/92% recall/96.4% specificity (when accuracy was considered the most important) and 94.1% accuracy/92.8% recall/95.2% specificity (considering recall). In the case of the KNN algorithm, there is one best model for both the recall and the accuracy, and its characteristics are: 94.3% accuracy/94% recall/94.5% specificity. In the cases of KNN and the best model based on bagged trees, for the accuracy, the subset of input features determined by the $\chi^{2}$ analysis was used. For the best bagged trees model considering recall, the method used to determine the input features was information gain.

White-box ML models put the focus on the interpretability of the models. The goal is to create a transparent process of prediction. They allow us to visualize what features were used in the prediction process and their behavior. To provide more information on the features suitable for the vulnerability prediction task, we decided to present the structure of the decision trees, which predict the occurrence of vulnerabilities on the basis of all features obtained (33 features) and let them decide what features should be used in a medium-sized model. In our analysis, we used fine trees; however, their structure is much larger, so for the purpose of the interpretation of the features used, we decided to build smaller models, which are more practical in this case. The accuracy of the models obtained was 75.9% (86.7% specificity, 62.6% recall) for the CWE-119 subset and 85.3% (81.6% specificity, 89.8% recall) for the CWE-399 subset.

In Figure 16, we present the structure of the tree trained on the subset with CWE-399 vulnerabilities considered. The feature comment_lines_density was used in the root of the tree. Furthermore, the parameter comment_lines is high in the hierarchy. This can suggest that the code might be complicated and needs many comments to be understood by the programmers. Many comments can also signal inadequacies in the code, but also the maturity of the code. Furthermore, features from the issue category (minor_issues and major_issues) are important for the prediction. They determine a number of potential problems in the code; here, the power of static code analyzer rules is used. Maintainability features (technical debt and code smells) are also used. The low maintainability and complexity of the code result in fault-proneness (proven in an empirical investigation [78]).

In Figure 17, a structure of the tree is presented, which was trained on the subset with CWE-119 vulnerabilities considered. In the root of the tree, one of the maintainability features can be found: code smells. Here, again, we can notice that the high maintainability and complexity of code can be a sign to consider the fault-proneness of the code [78]. Other features used in this model, functions, LOCCOM, complexity, and cognitive_complexity, also suggest that the size and complexity of the code can be indicators of software vulnerabilities. A number of functions (with a smaller number of functions, a probability of fault-proneness is bigger) can be a sign that large functions are used instead of small ones, and the single responsibility principle can be broken (but it is also strongly dependent on the size of the code element). The number of duplicated_lines, on the other hand, can indicate bad coding practices.

## 5. Conclusions

The aim of this work was to deliver a comprehensive evaluation of the features generated using static code analyzers for the purpose of vulnerability prediction and the delivery of guidelines considering the features that can be used as indicators of software vulnerabilities. Static code analyzers are often used at software companies to increase software quality, and works evaluating these tools are available in the literature, but focus on the usability of these tools and not on the discriminant power of the obtained metrics for the prediction of vulnerabilities. None of the works were focused on the feature selection regarding C/C++ languages, which are commonly used to build critical applications, operating systems, and virtual machines. What is more, our work delivers the most comprehensive feature analysis and selection (13 ML models, 33 features from static code analyzers, three correlation types, and four well-known feature selection techniques used). The results of this work aim to highlight the lack of analysis considering feature selection for C/C++ vulnerability prediction. What is more, our target is to inspire future works of the academic and industrial communities by delivering an extensive knowledge base and discussion considering a multitude of feature selection methods and their suitability for vulnerability prediction evaluated using a variety of standard and ensemble machine learning models.

The results of the correlation analysis show that the correlations between the features gathered from the static code analyzers are statistically significant in the majority of cases. The fact that for three out of five features from the CCCC tool, the hypothesis about the correlation significance was rejected suggests that it is better to use SonarQube to obtain the static analysis based features because it delivers a bigger number of features, which is more reliable in terms of statistical significance. This can be justified by the fact that SonarQube, in contrast to CCCC, is a commercial tool for static analysis and also outputs the metrics connected to the number of issues found in the code and is not limited to the traditional software metrics.

All the subsets of features examined in this work can be used to successfully train the ML classifiers. The analysis delivered us much information about the performance of different ML models on different subsets of data. Using the information gathered in the figures, one can choose the most suitable model for the problem and then check what subset of features is the most suitable for this particular model.

It was shown that the accuracy metric is not sufficient to evaluate the performance of the ML models. The aggregated results obtained for the different types of feature subsets show that although the accuracy results are comparable, the main difference lies in the recall and specificity, and here, a trade-off has to be considered. Early indication needs a higher specificity value and less false alarms. As an alternative, a more sensitive model can be created, which can result in more security alerts, but for which there is a bigger chance that the true vulnerability will be detected.

All of the methods (the correlation analysis, different ranking techniques, and the evaluation using different machine learning models) show that the CWE-399 vulnerabilities are the most statistically dependent on the features generated by a static code analyzer. These are resource management vulnerabilities. From that, we can infer that the difficulty of predicting the occurrence of the vulnerability is strongly dependent on its type.

Using different feature selection methods (correlation analysis, entropy based, chi-squared) and the interpretation of the white-box ML models (in our case, decision trees), we can determine the features that are the strongest indicators of vulnerabilities for the case of CWE-119 and CWE-399 vulnerabilities for C/C++ code elements. The results of our analysis show that features representing the size, character (density of comments, number of duplicated lines), and complexity of the code can be used for the purpose of vulnerability prediction. Furthermore, the maintainability metrics (code smells and technical debt) are determined by the techniques. These features can indicate bad coding practices, the fact that code may be over-complicated, etc. Furthermore, the low maintainability of the code can be a sign to perform security testing. SonarQube generates metrics considering the number of potential issues found in the examined code; these features should naturally be used to evaluate the security of the code.

Additionally, from our experiments, it appears that it is reasonable to create efficient ML models based on the features generated by static code analyzers, especially models created to predict the CWE-399 vulnerability type. To create the models considering only a particular type of vulnerabilities, it would be beneficial to evaluate the models on different subsets of features and consider only the best-ranked features determined for this particular class of vulnerabilities. This way, considering multiple types of vulnerabilities, multiple binary models could be created based on different subsets of input features.

## Figures and Tables

**Figure 1 sensors-21-01133-f001:**
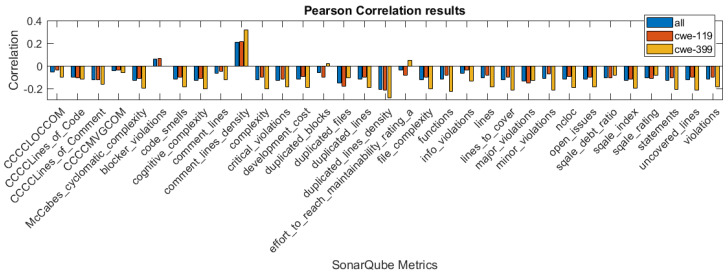
Pearson correlation values for the features obtained using different subsets of a dataset.

**Figure 2 sensors-21-01133-f002:**
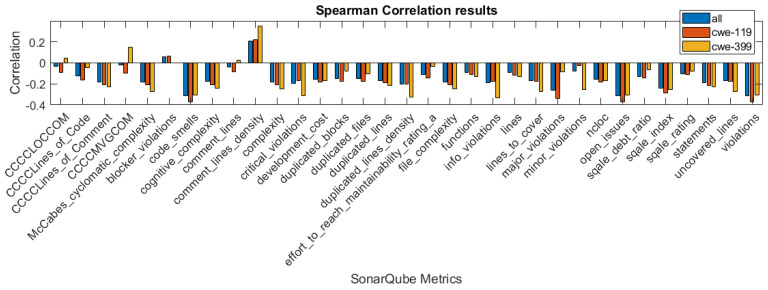
Spearman correlation values for the features obtained using different subsets of a dataset.

**Figure 3 sensors-21-01133-f003:**
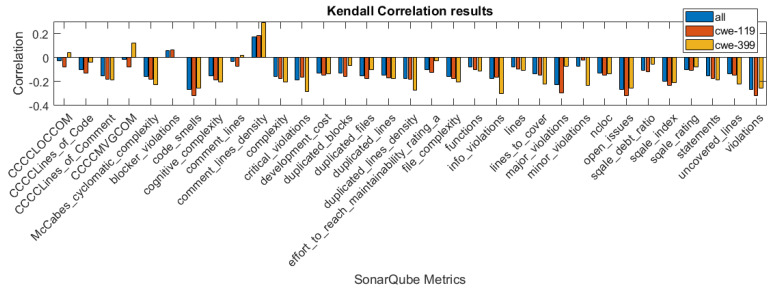
Kendall correlation values for the features obtained using different subsets of a dataset.

**Figure 4 sensors-21-01133-f004:**
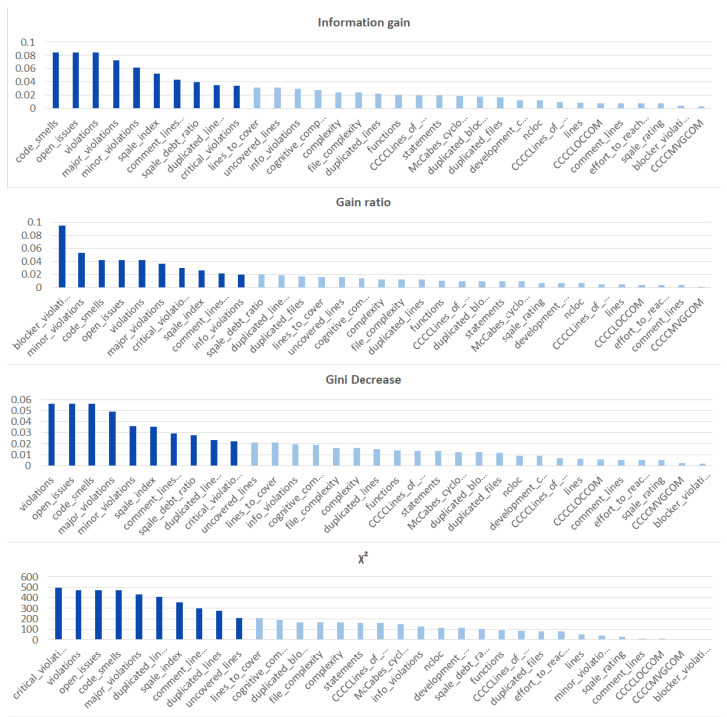
Feature ranks obtained using different types of metrics for the whole dataset.

**Figure 5 sensors-21-01133-f005:**
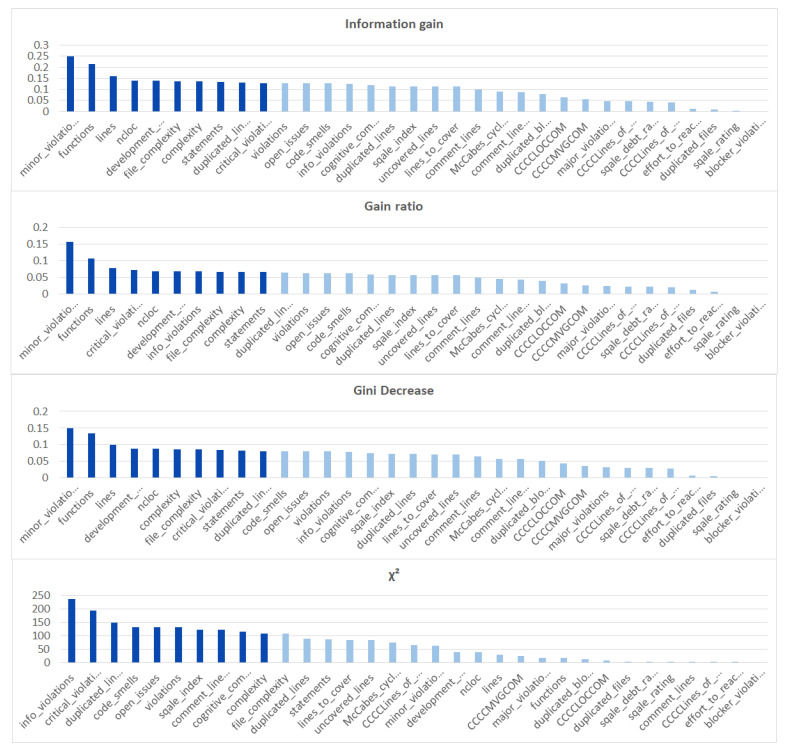
Feature ranks obtained using different types of metrics for the CWE-399 subset of the dataset.

**Figure 6 sensors-21-01133-f006:**
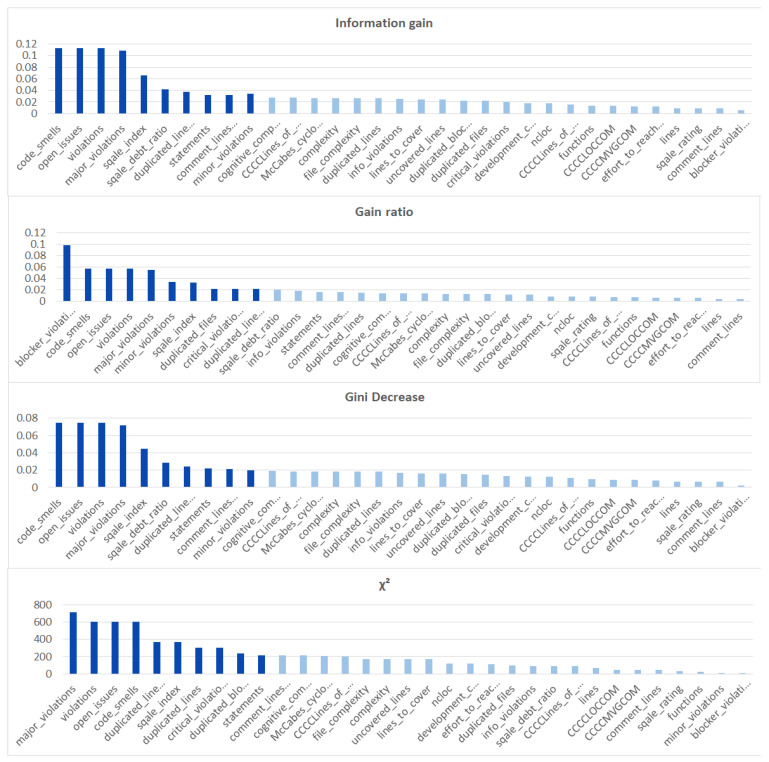
Feature ranks obtained using different types of metrics for the CWE-119 subset of the dataset.

**Figure 7 sensors-21-01133-f007:**
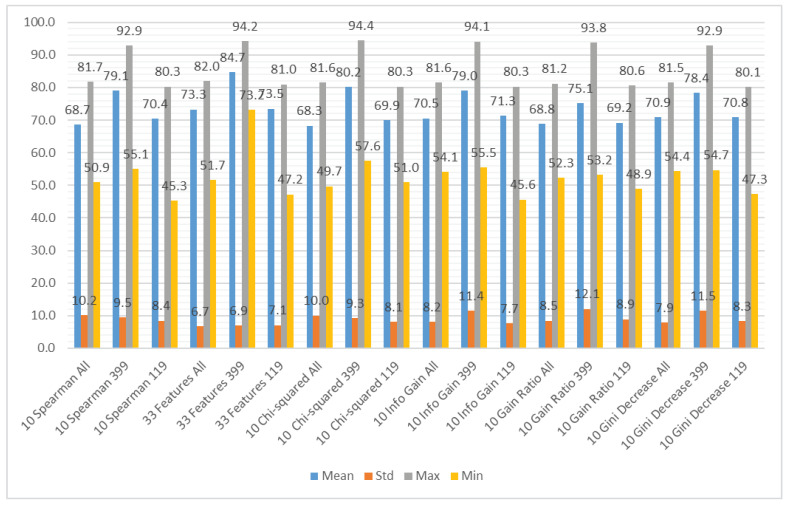
Aggregated accuracy results grouped by feature subset.

**Figure 8 sensors-21-01133-f008:**
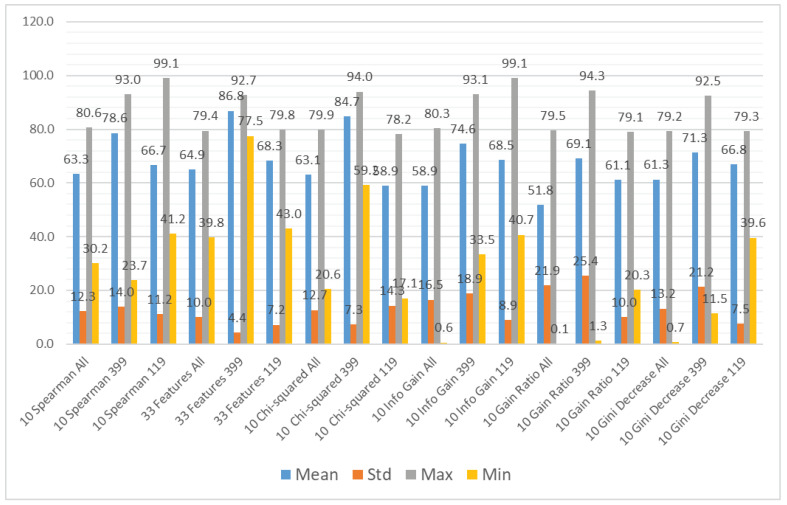
Aggregated recall results grouped by feature subset.

**Figure 9 sensors-21-01133-f009:**
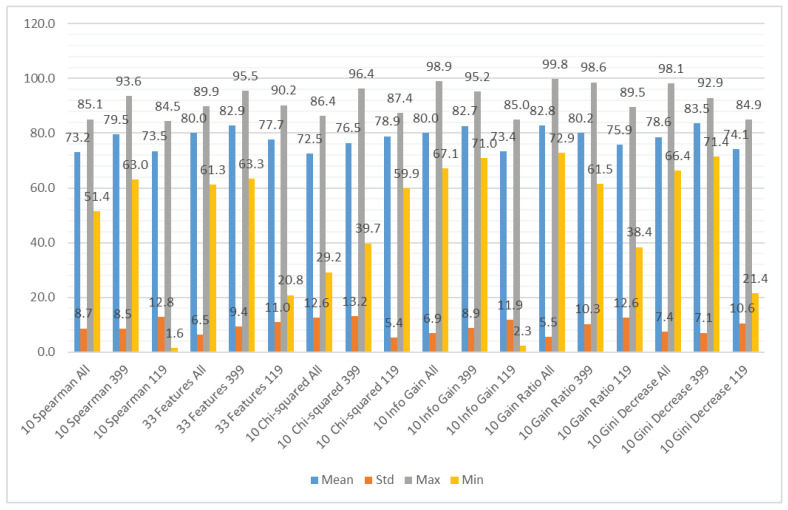
Aggregated specificity results grouped by feature subset.

**Figure 10 sensors-21-01133-f010:**
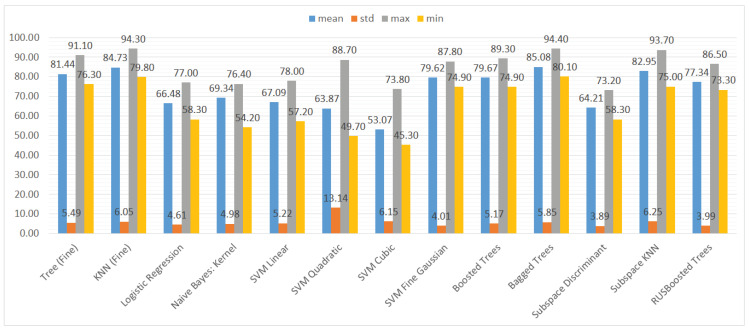
Mean, standard deviation, maximum, and minimum of the accuracy results for different types of classifiers.

**Figure 11 sensors-21-01133-f011:**
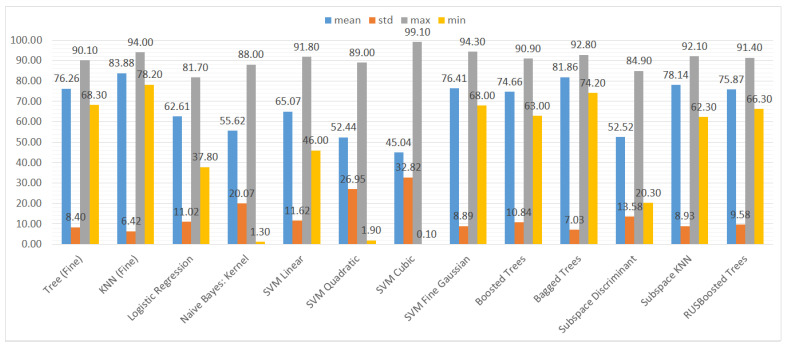
Mean, standard deviation, maximum, and minimum of the recall (TPR) results for different types of classifiers.

**Figure 12 sensors-21-01133-f012:**
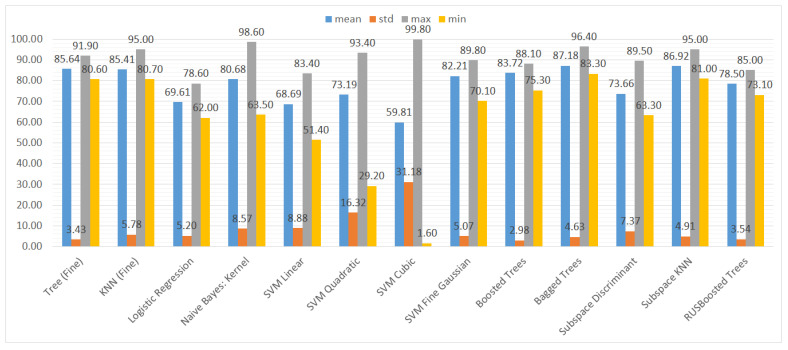
Mean, standard deviation, maximum, and minimum of the specificity (TNR) results for different types of classifiers.

**Figure 13 sensors-21-01133-f013:**
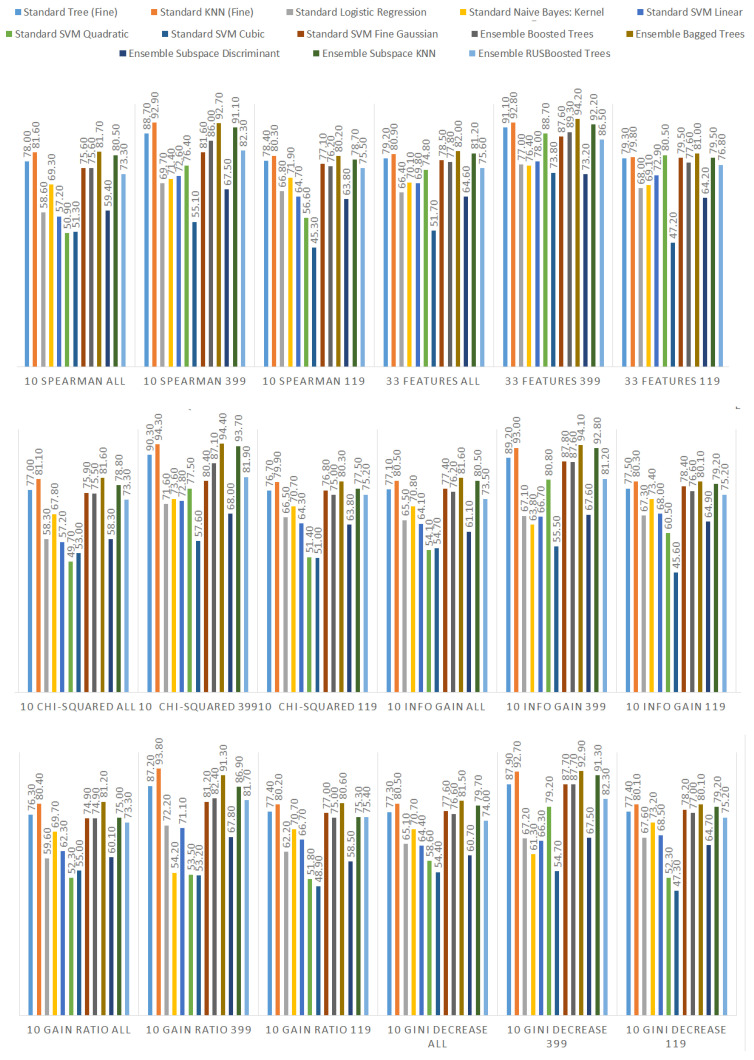
Detailed accuracy results for different types of classifiers.

**Figure 14 sensors-21-01133-f014:**
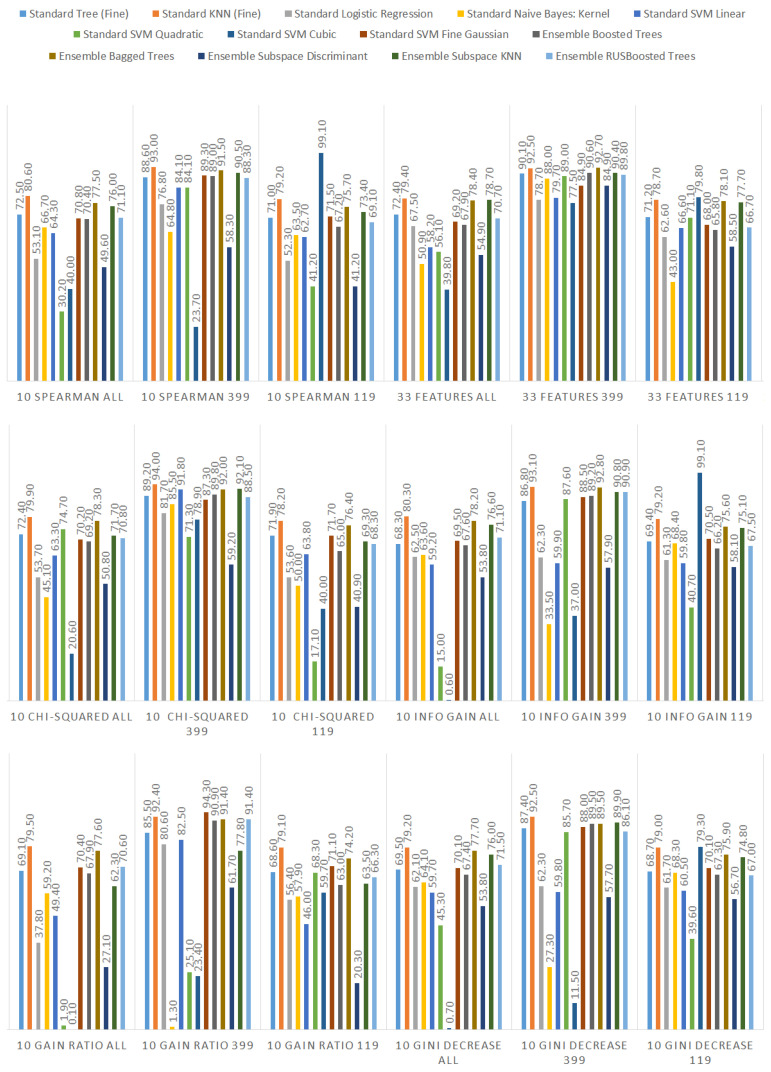
Detailed recall (TPR) results for different types of classifiers.

**Figure 15 sensors-21-01133-f015:**
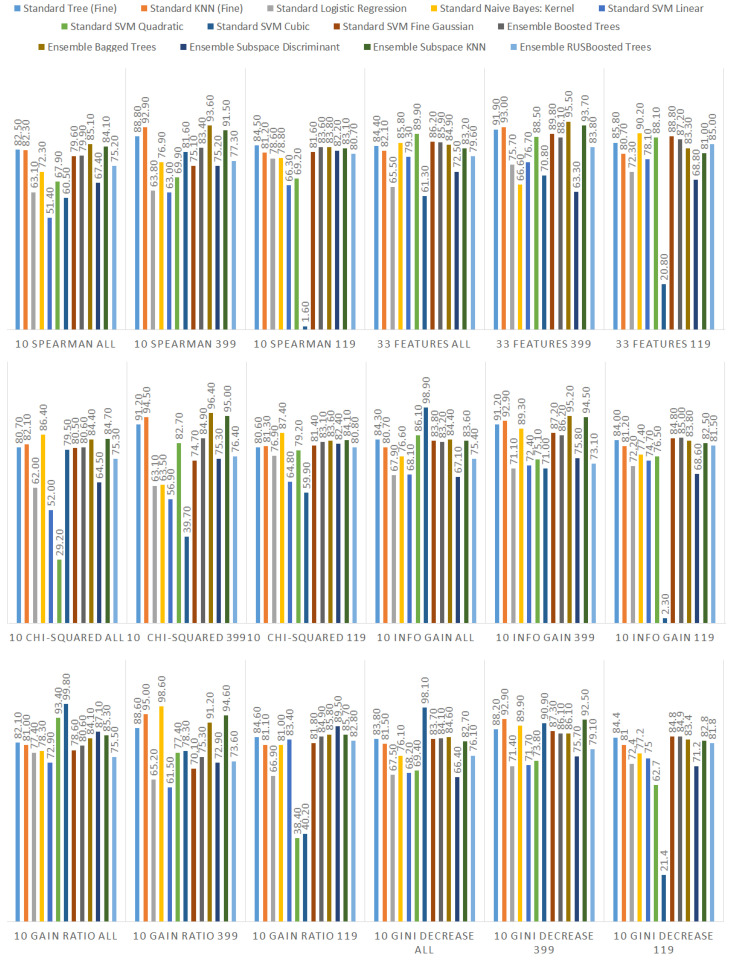
Detailed specificity (TNR) results for different types of classifiers.

**Figure 16 sensors-21-01133-f016:**
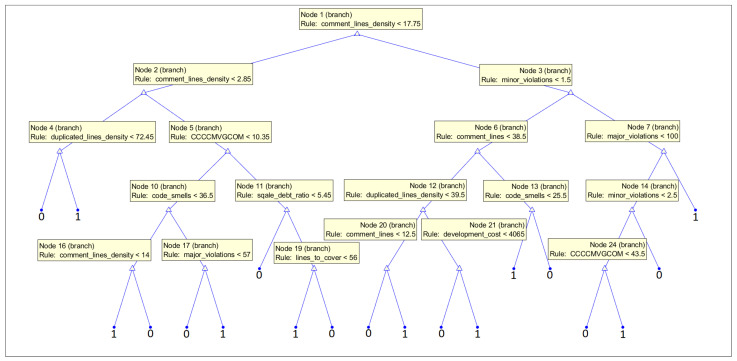
Medium tree based given all 33 features as the input for the subset with CWE-399 vulnerabilities considered.

**Figure 17 sensors-21-01133-f017:**
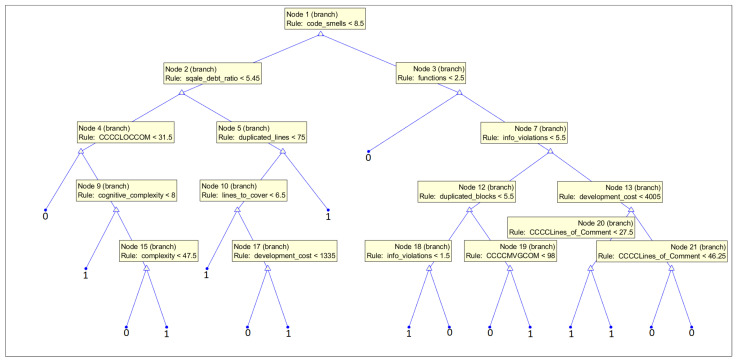
Medium tree based given all 33 features as an input for the subset with CWE-119 vulnerabilities considered.

**Table 1 sensors-21-01133-t001:** The cardinality of the code element sets with information on the class distribution used in the experiments. CWE, Common Weakness Enumeration.

Elements	Label	Cardinality	
All	Vulnerable	3386	7534
	Neutral	4148	
CWE-399	Vulnerable	684	1498
	Neutral	814	
CWE-119	Vulnerable	2702	6036
	Neutral	3334	

**Table 2 sensors-21-01133-t002:** *p*-values obtained in the correlation analysis for the columns, in which at least one *p*-value suggests that the null hypothesis should be rejected.

	Minor _Violations	CCCCLines _of_Code	Comment _Lines	Duplicated _Blocks	CCCCLOCCOM	CCCCMVGCOM	Effort_to_REACH _Maintainability_Rating_a
CWE-399							
Pearson	≪0.001	≪0.001	≪0.001	**0.4415**	0.0002	0.0231	**0.0513**
Spearman	≪0.001	**0.0726**	**0.3540**	0.0033	**0.0723**	≪0.001	**0.2308**
Kendall	≪0.001	**0.0726**	**0.3539**	0.00336	**0.0723**	≪0.001	**0.2307**
CWE-119							
Pearson	≪0.001	≪0.001	≪0.001	≪0.001	0.00439	0.0088	≪0.001
Spearman	**0.0694**	≪0.001	≪0.001	≪0.001	≪0.001	≪0.001	≪0.001
Kendall	**0.0694**	≪0.001	≪0.001	≪0.001	≪0.001	≪0.001	≪0.001
All							
Pearson	≪0.001	≪0.001	≪0.001	≪0.001	≪0.001	0.0009	0.0024
Spearman	≪0.001	≪0.001	0.0011	≪0.001	0.0027	**0.0581**	≪0.001
Kendall	≪0.001	≪0.001	0.0011	≪0.001	0.0027	**0.0581**	≪0.001

**Table 3 sensors-21-01133-t003:** Ten best features obtained for different ranking techniques.

Rank	Spearman Correlation	Information Gain	Gain Ratio	Gini Decrease	Chi-squared
1	code_smells	violations	blocker_violations	violations	critical_violations
2	open_issues	open_issues	minor_violations	open_issues	violations
3	violations	code_smells	violations	code_smells	open_issues
4	major_violations	major_violations	open_issues	major_violations	code_smells
5	sqale_index	minor_violations	code_smells	minor_violations	major_violations
6	comment_lines_density	sqale_index	major_violations	sqale_index	duplicated_lines_density
7	duplicated_lines_density	comment_lines_density	critical_violations	comment_lines_density	sqale_index
8	critical_violations	sqale_debt_ratio	sqale_index	sqale_debt_ratio	comment_lines_density
9	info_violations	duplicated_lines_density	comment_lines_density	duplicated_lines_density	duplicated_lines
10	statements	critical_violations	info_violations	critical_violations	uncovered_lines

## Data Availability

Data sharing not applicable

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
