# Peer review of "Efficient Feature Selection for Static Analysis Vulnerability Prediction"

_sensors, 2021, doi:10.3390/s21041133_

Round 1
Reviewer 1 Report
The authors examine features generated by SonarQube and CCCC tools, to identify features that can be used for Software Vulnerability Prediction. They investigate the suitability of thirty-three different features to train thirteen distinct Machine Learning algorithms. Feature selection process based on the correlation analysis of the features, together with four feature selection techniques
It is necessary to make two chapters Introduction and Related work from the existing chapter Introduction, because the chapter is a bit too long and confusing. Otherwise, the content of the introduction and related research are clearly presented.
The quality of figures is not satisfactory. The picture is overlapping the text in most of the figures. Please try to solve that problem.
Font in figure 10 is too small.
Implementation of several supervised machine learning methods aims at developing accurate and also interpretable models. Please can you give interpretation at least of some white – box type of machine learning models. It would be interesting to see some of these structured models and find out what domain knowledge is extracted from the dataset. Please do it at least for an illustration. A large number of machine learning procedures have been applied, so at least some models should be presented and an interpretation of interest to an expert in the field of application should be given.
Author Response
Thank you for the time and effort that you have devoted to analysing the content of our article and providing us with significant and constructive comments. All your comments allowed us to improve our paper.
We highlighted all the changes in pink (when a Figure was modified, a caption is pink). We improved the readability of Figures by increasing the font size and, in some cases, we merged subfigures into one figure (so as a result overlapping no longer occurs).
We also proofread the whole paper and removed the grammatical mistakes (we do not mark the linguistic modifications).
Also, we divided the Introduction into two sections: Introduction and Related Works. We hope that now the information is presented in a more orderly manner. Some adjustments were necessary here (e.g. in lines 129-131 and 135-153).
It was a great suggestion to add interpretation of the white-box ML models to find out what domain knowledge is extracted from the dataset. We expanded the article by performing additional experiments with medium-sized decision trees (which are smaller but easier to read and interpret than the ones used in the original experiments) for subsets with different types of vulnerabilities to increase its usefulness. [lines 366-373, 480-511 and Figures 13, 14]
Once more, we are grateful for your suggestions, which helped us to significantly improve our paper, and hope that the article will meet your demands.
Reviewer 2 Report
Factual issues:
- Referring to Symantec 2012 report [5] (even repeatedly in Intro) seem to be unnecessarily outdated.
- The “VP” abbreviation is used (page 4 top) without explicit explanation despite it seems almost obvious that it should stand for (Vulnerability Prediction)
- The reference to the dataset as [35] is not specific enough. The direct link to the dataset (e.g. https://github.com/CGCL-codes/VulDeePecker) should be used instead.
- The description of Pearson, Spearman and Kendall correlation coefficients (sect. 2.2, l. 242-248) is redundant and contains some inexact formulations. I suggest removing it and replacing with a suitable citation.
- The list of eight standard algorithms contains only seven of them (DT, KNN, LR, KNB and 3 SVM).
- Accuracy, Specificity and Recall are standard quantities used in classification tasks and they do not need to be defined again in the paper. Again, I suggest removing formulas (13)-(15) and replacing them with a suitable citation.
- Figures 1-3 are quite difficult to read I suggest merging all three subfigures in each of the Figures resulting in slightly better readable pictures. The similar is true for Figures 4-6.
- The section 2 Methodology does not describe the output obtained from Feature selection (2.2) and ML based evaluation specifically enough. Therefore, the experimental results described in section 3 and the conclusions seem to be insatifactory.
- In section 3, the results from various correlation coefficients and other metrics defined in sect. 2.2 are used rather arbitrarily not regarding their different nature and purpose. I suggest that more focus should be given to their differentiation in the application.
Formal issues:
- 1 legend for (b) seems to be partly covered by the edge of (c) image.
- The diagram in Fig. 13, 14 and 15 are so small that is renders unreadable.
English: fair, some errors/mistakes are present, e.g.
- “which” is used many times where “that” should have been preferred (e.g. lines 109, 110, 223, …).
- Use of non-existent word “automize” in line 223.
- Subsentence with something missing “takes into consideration the number of distinctive values the observed feature“ (below the line 261, above formula (10))
Author Response
Thank you for the time and effort that you have devoted to analysing the content of our article and providing us with significant and constructive comments. All your comments allowed us to improve our paper.
We highlighted all the changes in pink (when a Figure was modified, a caption is pink). We improved the readability of Figures by increasing the font size and, in some cases, we merged subfigures into one figure (so as a result overlapping no longer occurs).
We also proofread the whole paper and removed the grammatical mistakes (we do not mark the linguistic modifications).
We have improved our article by removing the outdated Symantec report and modifying the corresponding parts with information from newer sources [lines 28, 34-37]. We also added the explanation of the “VP” abbreviation [line 167]. We added the direct link to the dataset used in the study [lines 220, 142, 243]. We also added the missing ML algorithm name [line 359].
We removed a detailed description of Pearson, Spearman and Kendall correlation coefficients (sect. 2.2) and instead replaced it with suitable citations and more precise differentiation of these methods (outputs, purpose and nature) [lines 281-310]. We also expanded the differentiation of other feature selection methods used (entropy-based and chi-squared) [lines 311-319]. Also, the description of ML-based evaluation was changed [lines 348-355], the equations describing the standard ML metrics were replaced with a shorter description and a reference [lines 360-365]. The subsection was also expanded [lines 366-373]. We also extended the Experimental Results [lines 379-384, 480-511] and Conclusions [lines 557-568] sections to supplement these new parts and to meet your requirements. We hope that now, section Methodology, Experimental Results are complete and satisfactory.
Once more, we are grateful for your suggestions, which helped us to significantly improve our paper, and hope that the article will meet your demands.
Reviewer 3 Report
In this paper, the authors examine features generated by SonarQube and CCCC tools, to identify those that can be used for Software Vulnerability Prediction. The suitability of thirty-three different features is investigated to train thirteen distinct machine learning algorithms to design Vulnerability Predictors and identify the most relevant features that should be used for training.
The work is interesting and innovative in the domain of computer security. I have the two following concerns that need to be addressed before the paper can be accepted for publication:
- The authors mention that “In contrast to the related works, we analyze C/C++ applications. C/C++ languages are used in a variety of applications”. Thus, I am wondering how generalizable the work is to non-C/C++ applications. The authors need to discuss this point.
- Some recent state-of-the-art vulnerability analysis techniques that are overlooked in this paper such as:
- Resource-aware detection and defense system against multi-type attacks in the cloud: Repeated bayesian stackelberg game. IEEE Transactions on Dependable and Secure Computing(2019).
- Towards 5G-based IoT security analysis against Vo5G eavesdropping. Computing: 1-23.
The authors need to include these articles in the discussions.
Author Response
Thank you for the time and effort that you have devoted to analysing the content of our article and providing us with significant and constructive comments. Also, thank you for your kind comments! We are pleased that you find our work interesting and innovative.
We highlighted all the changes in pink (when a Figure was modified, a caption is pink). We proofread the whole paper and removed the grammatical mistakes (we do not mark the linguistic modifications).
As you suggested, we have added the explanation on how generalizable our work is to non-C/C++ applications. We hope that our explanation is sufficient. [lines 226-235]
We also extended the description of Vulnerability Analysis methods with the suggested papers. We hope that now the division of methods and the Related Work section (which was separated from the introduction) is now complete. [lines 174-184]
Once more, we are grateful for your suggestions, which helped us to significantly improve our paper, and hope that the article will meet your demands.
Round 2
Reviewer 1 Report
The authors have corrected paper according to suggestions. The paper is ready to be published.
Author Response
Thank you for your involvement in the process of improving our article and for your valuable suggestions! We are glad that our improvements have met your requirements! We are uploading the article with the editorial changes.
Reviewer 2 Report
Now, the paper is much better.
Just few rather formal issues:
- Paragraph starting in l. 174 ends only in line 213 and seems to be too long.
- Sentence in l. 296 – 298 contains an incorrectly formulated insert in parentheses (starts with “it” rather than “they”).
- 7 is alone in page 15
- In the term “vulnerability Prediction” in l. 525 “P” is capitalized with no apparent reason.
Author Response
Thank you for your involvement in the process of improving our article and for your valuable suggestions! We are glad that our improvements have met your requirements and that you think that our article is much better now! We have corrected all of the remaining issues:
line 298: we have changed an incorrectly formulated insert in parentheses: it does --> they do (highligthed (using pink))
lines 174-214: we have rebuilt the paragraph (which was too long), and as a result we divided it to 3 separate paragraphs (a few words were added to make it more natural - highligthed (using pink).
Figures 7-9: we have slightly minimised the figures to fit the first two of them on one page
line 526: we have corrected the wrongly capitalised letter
We are uploading the article with these changes.
Once more, thank you for your involvement and comprehensiveness!
Reviewer 3 Report
The authors have appropriately addressed my comments.
Author Response

(The authors gave the same response as above.)
